# Production, Characterization and Application of a Novel Chitosanase from Marine Bacterium *Bacillus paramycoides* BP-N07

**DOI:** 10.3390/foods12183350

**Published:** 2023-09-07

**Authors:** Yuhan Wang, Hongjuan Mo, Zhihong Hu, Bingjie Liu, Zhiqian Zhang, Yaowei Fang, Xiaoyue Hou, Shu Liu, Guang Yang

**Affiliations:** 1College of Food Science and Engineering, Jiangsu Ocean University, Lianyungang 222005, China; han85161995@126.com (Y.W.); mohongjuan73@163.com (H.M.); hzhwen621@126.com (Z.H.); 17851286193@163.com (B.L.); z18206289932@163.com (Z.Z.); foroei@163.com (Y.F.); 15822552894@163.com (X.H.); jdliushu@163.com (S.L.); 2Jiangsu Key Laboratory of Marine Bioresources and Environment, Jiangsu Ocean University, Lianyungang 222005, China; 3Co-Innovation Center of Jiangsu Marine Bio-Industry Technology, Jiangsu Ocean University, Lianyungang 222005, China; 4Jiangsu Key Laboratory of Marine Biotechology, Jiangsu Marine Resources Development Research Institute, Jiangsu Ocean University, Lianyungang 222005, China

**Keywords:** *Bacillus paramycoides*, chitosanase, chitosan, chitosan oligosaccharides, enzymatic properties

## Abstract

Chitooligosaccharides (COS), a high-value chitosan derivative, have many applications in food, pharmaceuticals, cosmetics and agriculture owing to their unique biological activities. Chitosanase, which catalyzes the hydrolysis of chitosan, can cleave β-1,4 linkages to produce COS. In this study, a chitosanase-producing *Bacillus paramycoides* BP-N07 was isolated from marine mud samples. The chitosanase enzyme (*Bp*CSN) activity was 2648.66 ± 20.45 U/mL at 52 h and was able to effectively degrade chitosan. The molecular weight of purified *Bp*CSN was approximately 37 kDa. The yield and enzyme activity of *Bp*CSN were 0.41 mg/mL and 8133.17 ± 47.83 U/mg, respectively. The optimum temperature and pH of *Bp*CSN were 50 °C and 6.0, respectively. The results of the high-performance liquid chromatography (HPLC) and thin-layer chromatography (TLC) of chitosan treated with *Bp*CSN for 3 h showed that it is an endo-chitosanase, and the main degradation products were chitobiose, chitotriose and chitotetraose. *Bp*CSN was used for the preparation of oligosaccharides: 1.0 mg enzyme converted 10.0 g chitosan with 2% acetic acid into oligosaccharides in 3 h at 50 °C. In summary, this paper reports that *Bp*CSN has wide adaptability to temperature and pH and high activity for hydrolyzing chitosan substrates. Thus, *Bp*CSN is a chitosan decomposer that can be used for producing chitooligosaccharides industrially.

## 1. Introduction

Chitin is the second largest biomass resource after cellulose, mainly found in the shells of shrimps and crabs and the cell walls of higher fungi or filamentous fungi [1,2]. The processing of shrimp, crab, and other aquatic products generates a large amount of the shrimp chaff waste every year, which is considered solid waste to be buried in soil. The processing and utilization of chitin can not only address environmental protection and sustainable development problems but also provide high-quality chitin-derived products [3,4]. Chitosan consists of N-acetylglucosamine and glucosamine (GlcN, D) linked by β-1,4-glucosidic bonds and is the only natural alkaline polysaccharide obtained by deacetylation of chitin [5]. Chitosan is widely distributed in nature and possesses diverse biological activities, but the high molecular weight and poor water solubility of chitosan limits its application and commercial development [6,7]. 

Chitooligosaccharide is an oligosaccharide product obtained by hydrolysis of chitosan and linked by 2–20 monosaccharides through a glycosidic bond [5,6]. It is a further-processed product of chitosan obtained by the deacetylation of chitin. Compared with chitosan, with its low molecular weight, good water solubility, easy absorbance by the human body and high biological activity, COS has outstanding performance in antimicrobial activity, tumor inhibition and regulation of plant immunity. COS has been approved in China as a new food resource, and has become another high-value derivative of chitosan after glucosamine [8]. Therefore, COS is attracting more and more attention and has broad application prospects. 

The traditional production of chitosan oligosaccharides mainly relies on physical and chemical methods, which have the disadvantages of high pollution, high energy consumption, high cost and low product purity [9]. The chitosan oligosaccharide used in this study was prepared by the chitosanase (E.C 3. 2.1. 132) hydrolysis method, which has the advantages of mild reaction conditions, uniform molecular weight distribution, high purity, and contributing to energy savings and environmental protection [10]. According to the Carbohydrate Active Enzyme Database (CAZY, http://www.cazy.org/, accessed on 11 July 2023) of the different sequences of amino acids, chitosanases can be classified into seven different families, namely GH-3, GH-5, GH-7, GH-8, GH-46, GH-75 and GH-80 [1,11]. Among these families, only GH-46, GH-75 and GH-80 include enzymes that specifically hydrolyze chitosan. Compared to other families of chitosanases, the GH-46 family has the widest range of members with chitosanase activity, the most extensive membership and are the most intensively studied chitosanases in terms of structure and function [12]. The catalytic site of GH-8 family chitosanases is located on the (α/α)_6_ barrel structure, which is composed of 6 α-helices forming a central barrel, the outer layer surrounds another 6 α-helices forming a barrel structure, and the six repeating helix–loop–helix motifs form a barrel structure by bilayer [13]. Fungal chitosanases are mainly distributed in the GH-75 family, largely *Aspergillus* [12]. According to the different mechanisms of chitosanase-catalyzed hydrolysis of chitosan, chitosanases can be divided into two categories. One is endo-chitosanase, which can hydrolyze chitosan to produce oligosaccharides with different DP, but cannot hydrolyze disaccharides, and the representative strain is *Bacillus* sp. DAU101 [14]. The other category is exo-chitosanase, which can only hydrolyze from one end of the chitosan molecular chain, cutting off one glucosamine at a time to produce monosaccharides, such as the representative strain *Thermococcus kodakararaensis* KOD1 [15]. 

Chitooligosaccharides not only retain some functional properties of chitosan but also have water-soluble and other more valuable characteristics. Enzymatic degradation of chitosan is a widely-used method; it can split β-1,4-glyeosidic bonds specifically and selectively [16]. The traditional chitosanase production method is obtained by the natural fermentation of wild bacteria. The fermentation conditions and forms of different strains are different, and the fermentation conditions of wild bacteria are immature. However, these enzymes have low activity, poor thermal stability and acid resistance, which cannot meet the demand for enzymatic preparation of chitooligosaccharides in industrial production [17]. Therefore, the discovery of new chitosanases with high catalytic activity has raised great interest among researchers, especially the enzymes with special properties such as those with good thermal stability or adaptability to low pH [18]. The chitooligosaccharide produced by chitosanases has excellent performance, the advantages of uniform molecular weight distribution, high purity, and no other byproducts [1]. In this study, to discover some new properties of excellent chitosanase for use in the large-scale production of COS, we screened a chitosanase-producing marine strain from sea mud samples from the Yanwei Port, Haizhou Bay, Lianyungang City, Jiangsu Province. The production and partial characterization of chitosanase *Bp*CSN from a newly isolated *Bacillus paramycoides* BP-N07 were evaluated. The results will make a positive contribution to chitosan industrial technology upgrade and quality improvement.

## 2. Materials and Methods

### 2.1. Materials and Reagents 

Chitosan (the degree of deacetylation, DD ≥ 98.0%) was purchased from Shanghai Aladdin Bio-Chem Technology Co., LTD (Shanghai, China). Various chitosan–oligosaccharide standards (GlcN, (GlcN)_2_, (GlcN)_3_, (GlcN)_4_, (GlcN)_5_, (GlcN)_6_ and, (GlcN)_7_) were purchased from Qingdao Bozhi Huili Biotechnology Co., Ltd. (Qingdao, China). The preparation of colloidal chitosan involves dissolving 1.0 g of chitosan in 100 mL of deionized water, then adding acetic acid solution to dissolve it and adjusting the pH to 6.0 with sodium acetate. The silica gel plate for thin-layer chromatography (TLC) was obtained from Merck Chemicals Co., Ltd. (Shanghai, China). The synthesis of primers, DNA sequencing and biological kits were performed or purchased from Tsingke Biotechnology Co., Ltd. (Qingdao, China). All other reagents and solvents (>98% purity) were obtained from Sangon Biotech Co., Ltd. (Shanghai, China). 

### 2.2. Isolation and Identification of Chitosanase-Producing Strains

Sea mud samples for screening of chitosanase-producing strains were obtained from Yanwei Port, Haizhou Bay, Lianyungang City, Jiangsu Province. One gram of sea mud samples was weighed and cultured in a 50 mL enrichment medium with 100 μg/mL of ampicillin (KH_2_PO_4_ 0.1%, K_2_HPO_4_·3H_2_O 0.2%, MgSO_4_·7H_2_O 0.07%, NaCl 0.1%, chitosan 0.25%, yeast extract 0.25%, pH 6.0). After enrichment, the suspensions of enriched samples were diluted and again spread onto a selective culture medium with 100 μg/mL of ampicillin (same as above) to preliminarily assess the enzyme activity based on the degree of transparency and to determine the enzyme activity. The strain with the highest enzyme activity was used in this study. It was identified by morphological observation and Gram staining. The 16S rDNA sequences of obtained strain were amplified by PCR using the primers (27F: 5′-AGAGTTTGATCCTGGCTCAG-3′ and 1492R: 5′-TACGGCTACCTTGTTACGACTT-3′), and verified by DNA sequencing. The sequencing results were submitted to GenBank with Accession No. OQ930621.1. The phylogenetic analysis of 16S rDNA from different strains was generated using MEGA 11.0.

### 2.3. Determination of Chitosanase Activity

The enzyme activity of chitosanase was determined by the DNS (3,5-dinitrosalicylic acid) method, and different concentrations of glucosamine (0.1–1.0 mg/mL) were used to prepare standard curves (Appendix A) [19]. The reaction mixture (1 mL) consisted of 0.9 mL colloidal chitosan and 0.1 mL enzyme solution. The enzymatic reaction was carried out in a 50 °C water bath for 15 min, and the reaction was terminated in a boiling water bath for 10 min. After centrifugation, 0.1 mL of the supernatant was poured into a new EP tube with 0.2 mL of DNS solution and boiled in a water bath for 5 min. Then, 0.9 mL of distilled water was added, and 200 μL were placed in a 96-well plate and measured at 540 nm. The enzyme activity unit (U) is defined as the amount of enzyme required to catalyze the production of 1 μmol of reducing sugars per minute at 50 °C, and the specific activity is defined as mg^−1^ chitosanase unit.

### 2.4. Determination of the Growth and Enzyme Production Curves of Bacillus paramycoides BP-N07

The single colony on the plate was scraped with a disposable inoculation loop and plugged into the seed medium for activation. After overnight activation, a 2% seed solution was added into 200 mL of the fermentation medium (soluble chitosan 1.0%, (NH_4_)_2_SO_4_ 1.0%, K_2_HPO_4_·3H_2_O 0.14%, NaCl 0.5%, MgSO_4_·7H_2_O 0.13%, KH_2_PO_4_ 0.03%, yeast extract 0.3% and pH 5.0) and cultured at 30 °C. The absorbance value was measured every 2 h under OD_600_ conditions to plot the growth curve. Meanwhile, the enzyme activity of chitosanase was determined in the supernatant of the fermentation medium every 2 h to plot the enzyme production curve.

### 2.5. Production and Purification of BpCSN 

*Bacillus paramycoides* BP-N07 was inoculated into the fermentation medium at 30 °C and centrifuged at 180 rpm/min for 2 days, and the supernatant was collected for purification using ammonium sulfate precipitation. Then, the enzyme was purified by ion-exchange chromatography and gel filtration chromatography. The peak activity of the protein and the target protein peak eluent were collected, then desalted and concentrated. The desalted samples were analyzed by 12% sodium dodecyl sulfate polyacrylamide gel electrophoresis (SDS-PAGE), and the protein bands were stained with Coomassie Brilliant Blue G-250 at the end of SDS-PAGE. The concentration of proteins was determined using the BCA protein assays as described in [4].

### 2.6. Zymogram Analysis of Chitosanase Enzyme BpCSN

The zymogram analysis of *Bp*CSN was performed using the Congo Red method described by Bergmeyer [20] with modification. An amount of 0.1 g/100 mL of chitosan was added to the SDS-PAGE separation gel, and the electrophoresis was complete. After completion, the separation gel was added to the refolding buffer (50 mmol/L KH_2_PO_4_-NaOH (PH 5.5), 1 g/100 mL Triton X-100) and soaked for 10–12 h. Then, the separating gel was incubated in 10 mmol/L acetic acid–sodium acetate solution (pH 5.5) at 45 °C for 4 h. The chitosanase activity on the gel was visualized by 0.1 g/100 mL Congo red staining for 20 min, and 1 mol/L NaCl solution decolorized until the band was clear. 

### 2.7. Determination of Optimal pH and pH Stability of BpCSN

The optimal pH of *Bp*CSN was determined from 50 mM buffers at 50 °C with pH values between 3.0 and 9.0, namely an acetate buffer (pH 3.0–6.0), phosphate buffer (pH 6.0–8.0) and Tris-HCl buffer (pH 8.0–9.0). All enzymatic assays under different conditions were incubated for 15 min to determine the remaining activity. The pH stability of *Bp*CSN was evaluated by first preincubating the enzyme solution in 50 mM buffers at different pH values (pH 3.0–9.0) at 4 °C for 24 h. The residual activities were analyzed immediately under standard conditions (optimal pH, 50 °C for 15 min). The original activity at optimal pH (6.0) was considered 100%, and the percentage of the residual activity at different pH values against the original one at optimal pH (6.0) was calculated.

### 2.8. Determination of Optimal Temperature and Thermal Stability of BpCSN

The optimal temperature of *Bp*CSN was determined in 50 mM acetate buffer (pH 6.0) between 10 and 90 °C, and all enzymatic reactions at different temperatures were incubated for 15 min, then inactivated in a boiling water bath for 10 min to determine the remaining activity. In order to explore the thermal stability of *BpCSN*, the enzyme solution was placed in a water bath at 30 °C, 40 °C, 50 °C, 60 °C, and 70 °C for 4 h, 8 h, 12 h and 16 h, respectively. The residual activities were measured under standard conditions. The original activity at pH 6.0 was considered 100%, and the percentage of the residual activity at different time points and temperatures against the initial one was calculated.

### 2.9. Effects of Metal Ions on Enzyme Activity of BpCSN

To investigate the effect of metal ions (Fe^3+^, Ca^2+^, Mg^2+^, Zn^2+^, Mn^2+^, Ba^2+^, Pb^2+^, Hg^2+^, Ag^+^) on the activity of chitosanase *Bp*CSN, a solution of these metal ions with a concentration of 2 mM was prepared. An enzyme solution was incubated in the metal ion solution at 4 °C for 1 h, and then the catalytic activity with H_2_O was used as a control (100% relative activity). Two groups of chitosanases reacted with the 1% (*w/v*) substrate (pH 6.0) for 15 min to determine the residual activity of the enzyme.

### 2.10. Determination of Kinetic Parameters and Substrate Specificity of BpCSN

The substrate specificity of *Bp*CSN was evaluated in 15 min reactions at 50 °C in 50 mM buffer with different substrates (2% *w/v*), including colloidal chitosan, insoluble chitosan, insoluble chitin, and water-soluble chitosan. Enzyme activity was measured by the DNS method, with the highest activity being 100%.

The kinetic parameters of *Bp*CSN were analyzed with different concentrations (3–10 mg/mL) of colloidal chitosan substrates. The reaction was carried out in 0.2 M sodium acetate buffer (pH 6.0) at 30 °C for 15 min. The kinetic parameters *K*m and *V*max of the reaction between enzyme and substrate were calculated using Lineweaver–Burk plots.

### 2.11. Analysis of the Hydrolytic Products of BpCSN by TLC and HPLC

The hydrolysis products of *Bp*CSN were analyzed using colloidal chitosan as the substrate. The reaction mixtures containing 100 μL of chitosanase and 900 μL of 1% (*w/v*) substrate (pH 6.0) were incubated at 50 °C for 0–4 h. Samples were extracted at different times (0 min, 5 min, 10 min, 15 min, 30 min, 45 min, 60 min, 90 min, 2 h, 3 h, 4 h) and boiled immediately for 10 min. The degradation products were analyzed by thin-layer chromatography (TLC) as described by Mao et al. and high-performance liquid chromatography (HPLC) as described by Zhao et al. [21,22]. To further investigate the hydrolysis pattern of *Bp*CSN, chitotetraose and chitopentaose ((GlcN)_4_ and (GlcN)_5_) were treated with the enzyme for 2 h and the degradation products analyzed by TLC.

### 2.12. Statistical Analysis 

All the experiments were carried out independently in triplicate (*n* = 3), and one-way ANOVA and Duncan algorithm were used to test for variance and significant differences using SPSS 26.0 (SPSS Inc., Chicago, IL, USA).

## 3. Results 

### 3.1. Isolation and Identification of Chitosanase-Producing Strain

Chitosanase is the key enzyme for the green and efficient production of chitooligosaccharides. However, its low catalytic efficiency limits further research and application. In this study, a chitosanase-producing strain BP-N07 was screened with the highest enzyme activity and the biggest transparent circle size (Figure 1A), which was a Gram-positive bacterium with slimy white colonies and rod-shaped cellular morphology (Figure 1B,C). The 16S rDNA of strain BP-N07 was sequenced and submitted to GenBank with Accession No. OQ930621.1, and the phylogenetic tree was constructed and displayed using MEGA11.0 in Figure 1D. The phylogenetic analysis identified that strain BP-N07 was clustered together with *Bacillus paramycoides* and had a close genetic relationship, so the strain BP-N07 was identified as *Bacillus paramycoides*. 

### 3.2. Time-Course Expression Profile of Chitosanase by Bacillus paramycoides BP-N07

To obtain the time-course expression profile of chitosanase by *B. paramycoides* BP-N07 (*Bp*CSN), we determined the growth curve (Figure 2A) and enzyme production curve (Figure 2B) in the fermentation medium. According to the results shown in Figure 2, the expression of chitosanase by *B. paramycoides* BP-N07 is growth-associated since an increase in cell density is concomitant with an increase in chitosanase activity but not synchronized. After culturing for 2–12 h, *B. paramycoides* BP-N07 grew in the logarithmic stage, with a sharp increase in biomass, but the enzyme activity of chitosanase was barely detectable. After incubation for 12 h, the growth of *B. paramycoides* BP-N07 entered a period of stabilization with OD_600_ values maintained between 1.6 and 1.8. Meanwhile, extracellular chitosanase activity increased sharply throughout the culture period and the highest enzyme activity reached 2648.66 ± 20.45 U/mL at 52 h.

### 3.3. Purification and Zymogram Analysis of BpCSN

Ammonium sulfate precipitation, ion-exchange chromatography and gel filtration chromatography were used for the purification of *Bp*CSN. SDS–PAGE analysis followed by in-gel activity staining and Coomassie Brilliant Blue G-250 showed that the molecular weight of purified *Bp*CSN was approximately 37 kDa, which was the specific protein bands with chitosanase enzyme activity in the zymogram analysis (Figure 3). The yield and enzyme activity of *Bp*CSN were 0.41 mg/mL and 8133.17 ± 47.83 U/mg, respectively. Interestingly, after induction with 2.0% insoluble chitosan, the protein concentration of *Bp*CSN increased in the fermentation broth compared to the 1.0% insoluble chitosan inducer, while the background protein was reduced.

### 3.4. Effect of pH and Temperature on the Catalytic Activity of BpCSN

As shown in Figure 4A, no activities of *Bp*CSN were detected below pH 3.0 nor above pH 9.0 (data not shown), while the optimal pH was 6.0. Meanwhile, *Bp*CSN retained residual activity above 80% between pH 5.0 and pH 8.0. The pH stability of *Bp*CSN was investigated after incubation for 22 h in the pH range of 3.0–9.0 (Figure 4B). The relative enzyme activity of *Bp*CSN was above 80% at pH 3.0–9.0, which had excellent pH stability.

The optimal temperature and thermal stability were determined (Figure 4C,D). The optimal reaction temperature for *Bp*CSN was 50 °C and less than 80% of residual activity was at 40–80 °C. The catalytic activity of *Bp*CSN was very stable at 30–50 °C and retained over 80% of residual activity after incubation for 16 h at pH 6.0. Therefore, *Bp*CSN is a heat-resistant chitosanase.

### 3.5. Effect of Metal Ions on Enzyme Activity of BpCSN

At the same time, we also detected the effect of metal ions on the activity of *Bp*CSN (Figure 5). The results reveal that the majority of metal ions do not affect the enzymatic activity of *Bp*CSN. Only Mn^2+^ can upregulate the activity of *Bp*CSN by 25.8%, and downregulate by 48.7%, 26.4% and 37.2% in the presence of Fe^3+^, Ag^2+^ and Hg^2+^, respectively.

### 3.6. Determination of Substrate Specificity and Kinetic Parameters of BpCSN

As shown in Figure 6, the substrate specificity of *Bp*CSN was determined. *Bp*CSN towards colloidal chitosan had the highest specific activity, followed by soluble chitosan. However, *BpCSN* can hardly enzymatically degrade colloidal chitin, insoluble chitin and insoluble chitosan to produce chitooligosaccharides. Furthermore, the kinetic parameters of *Bp*CSN on colloidal chitosan were determined by substrate hydrolysis from 0.3 to 1.0% (*w/v*) for 15 min. The apparent *K*m and *V*max values of *Bp*CSN for chitosan were 4.063 ± 0.288 mg/mL and 2727.03 ± 192.97 µmol/min/mg, respectively (Appendix A).

### 3.7. Identification of Hydrolysis Product and Investigation of Hydrolysis Patterns of Chitosan Catalyzed by BpCSN

The hydrolytic process of *Bp*CSN on colloidal chitosan was detected by thin-layer chromatography. As shown in Figure 7A, from the mixture of COSs including (GlcN)_2_ to (GlcN)_7_ in the initial stage of the reaction and after 1 h, the substrate was mostly degraded to low DP COSs. As the reaction proceeded, TLC analysis after 3 h showed that the main degradation products by *Bp*CSN were chitobiose, chitotriose and chitotetraose ((GlcN)_2_, (GlcN)_3_ and (GlcN)_4_), but no monomeric (GlcN) was observed, indicating that it is an endo-interacting enzyme. In order to understand the details of the hydrolysis mechanism, the hydrolysis patterns of chitotetraose and chitopentaose ((GlcN)_5_) were further investigated by TLC. The results show that the *Bp*CSN had difficulty in hydrolyzing chitotriose but could effectively catalyze chitopentaose production of chitobiose and chitotriose (Figure 7B). HPLC results of colloidal chitosan treated with *Bp*CSN for 3 h further confirmed the above results (Appendix A). Finally, *Bp*CSN was used for the preparation of oligosaccharides: 1.0 mg enzyme converted 10.0 g chitosan with 2% acetic acid into oligosaccharides at 50 °C there was no chitosan in the reaction solution after 3 h (Appendix A).

## 4. Discussion

Chitooligosaccharide has excellent properties such as antibacterial, anti-tumor and moisturizing [10], and its safety and biocompatibility have great potential in clinical application [23]. The antioxidant and immunostimulant activities of chitosan oligosaccharide make it superior to chitosan in the food industry, not only maintaining food quality and prolonging shelf life, but also being potential antioxidants and valuable food additives [24]. COS can be obtained by hydrolyzing chitosan with chemical or biological enzymes. Chitosanases enzymatic hydrolysis deals with chitosan to efficiently produce COS. The whole process can reduce emissions of strong alkalis, save large amounts of washing water, ensure high production and highly active enzymes, prepare functional activity of oligosaccharides, greatly reduce the reaction time in the process of preparation, cut down the consumption of chemical reagents, and reduce costs, conforming to the requirements of environmental protection [22]. The low thermostability and activity of enzymes are the bottlenecks for their application at an industrial scale. Following the wide application of chitooligosaccharide, there is an urgent need for high thermostable and active chitosanase.

In this study, a novel chitosanase *Bp*CSN from the marine bacterium *Bacillus paramycoides* BP-N07 was isolated. The molecular weight of most chitosanases is usually approximately 20–50 kDa, with a few fungal chitosanases having molecular weights of approximately 100 kDa (Appendix A) [1,14,21,25,26,27,28,29,30,31,32,33]. Here, the molecular weight of purified *Bp*CSN was approximately 37 kDa, which was not different from the chitosanases of most microorganisms. The optimum temperature and pH of *Bp*CSN were 50 °C and 6.0, respectively. The relative enzyme activity of *Bp*CSN was above 80% at pHs 3.0–9.0 or 30 °C–60 °C. *Bp*CSN is an acidic chitosanase with excellent pH and thermal stabilities. *Bp*CSN with specific enzymatic properties tends to have a better advantage in chitosan substrate hydrolysis of industrial applications. Since chitosan shows better solubility at higher temperatures, the heat-resistant chitosanase not only maintains high catalytic activity under these reaction conditions but also reduces the inhibition of hydrolysis due to the high viscosity of the solution [34,35]. Unlike pH-resistant chitosanases, *Bp*CSN can hydrolyze the substrate at low pH, which significantly reduces the side effects caused by the Meladic reaction [36]. Chitosan can be efficiently converted to COS by chitosanase *Bp*CSN. Colloidal chitosan is first randomly cleaved into GlcN oligomers during the enzymatic reaction for 3 h, followed by further formation of (GlcN)_2_–(GlcN)_4_, which is an endo-interacting enzyme and has similarities with chitosanases from *Bacillus* sp. (Appendix A) [1,14,21,25,26,27,28,29,30,31,32,33]. Production efficiency is also important from an industrial point of view, as short periods can significantly reduce energy consumption and production costs. It is noteworthy that the reaction time in this study (3 h) was much shorter than the required reaction time (6–68 h) reported in previous studies [37]. Meanwhile, *Bp*CSN can catalyze the cleavage of the β-1,4-glycosidic bonds in chitosan but cannot catalyze chitin, which showed a higher hydrolysis capacity for highly deacetylated colloidal chitosan and water-soluble chitosan but no catalytic activity towards low deacetylated chitosan. Kinetic analysis of the hydrolysis of colloidal chitosan by *Bp*CSN showed that *Bp*CSN exhibited higher catalytic efficiency for long-chain substrates.

At present, the majority of chitosanases discovered and extensively studied are normally endo-type enzymes, which hydrolyze chitosan through an endo-type pattern, yielding a mixture of oligosaccharides ranging from dimers to octamers [28,29,30,31,32,33,38,39]. *Bp*CSN can hydrolyze chitopentaose to chitobiose and chitotriose, which indicates that chitosanases can split the β-1,4-glycosidic linkages of GlcN–GlcN. However, the chitosanase was unable to further hydrolyze chitotriose and chitotetraose, and no monomeric glucose was produced [1]. Studying the crystal structures of *Bp*CSN is a suitable method to elucidate the molecular mechanism of efficient degradation of chitosan catalyzed by chitosanases, which, in the future, will be the next scientific research subject of our research group. In summary, these results indicate that *Bp*CSN from *Bacillus paramycoides* BP-N07 obtained by this experimental screening is suitable for the commercial production of chitosan hydrolysates such as chitosan oligosaccharides.

## 5. Conclusions

In this study, a chitosanase-producing strain of *Bacillus paramycoides* BP-N07 was screened, which was fermented in large quantities to prepare chitosanase *Bp*CSN with highly active and stable chitosanase. The optimum temperature and pH of *Bp*CSN were 50 °C and 6.0, respectively. The main hydrolysis products of *Bp*CSN are (GlcN)_2_, (GlcN)_3_ and (GlcN)_4_. The implementation of this study would expand the resources of novel chitosanases that can be applied to prospective industrial needs. Our results deepen the understanding of the catalysis and property formation mechanisms of the chitosanase family. The heat-stable and high-energy chitosanase obtained in this study can reduce the production cost of shrimp and crab shell processing in China, which is of great significance for the effective utilization of chitin resources and improvement of the value of seafood byproducts.

## Figures and Tables

**Figure 1 foods-12-03350-f001:**
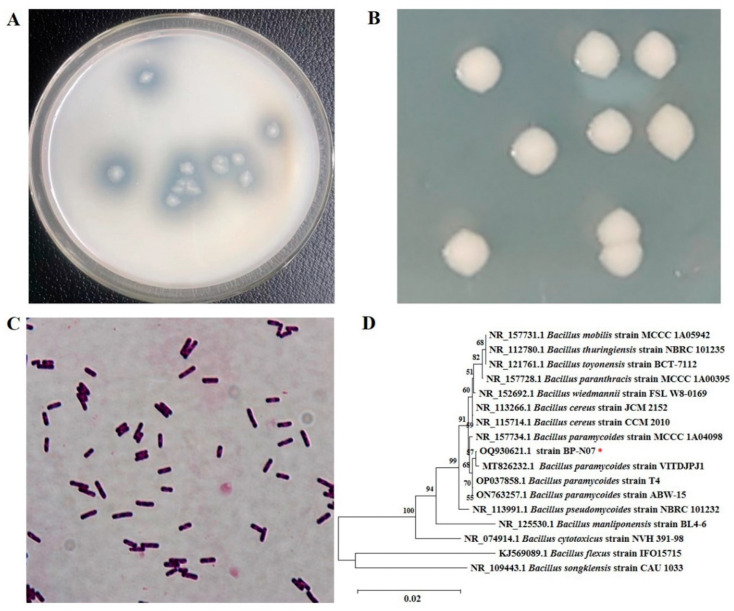
Isolation and characterization of *Bacillus paramycoides* BP-N07. (**A**) The transparent circle of *Bacillus paramycoides* BP-N07. (**B**) Colony morphology of *Bacillus paramycoides* BP-N07. (**C**) Gram staining of *Bacillus paramycoides* BP-N07. (**D**) The phylogenetic tree of 16S rDNA of *Bacillus paramycoides* BP-N07. The 16S rDNA of *Bacillus paramycoides* BP-N07 is indicated with the red asterisk.

**Figure 2 foods-12-03350-f002:**
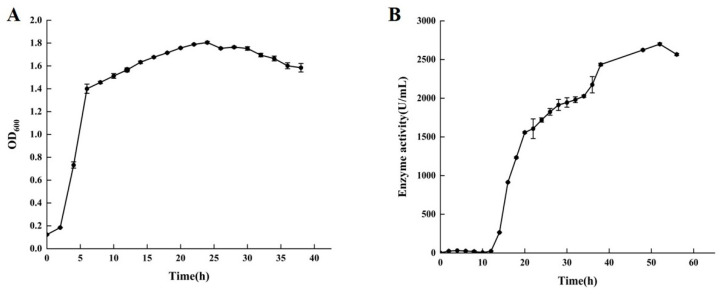
Time-course expression profile of chitosanase by *Bacillus paramycoides* BP-N07. (**A**) Growth curve. (**B**) Enzyme activity curve.

**Figure 3 foods-12-03350-f003:**
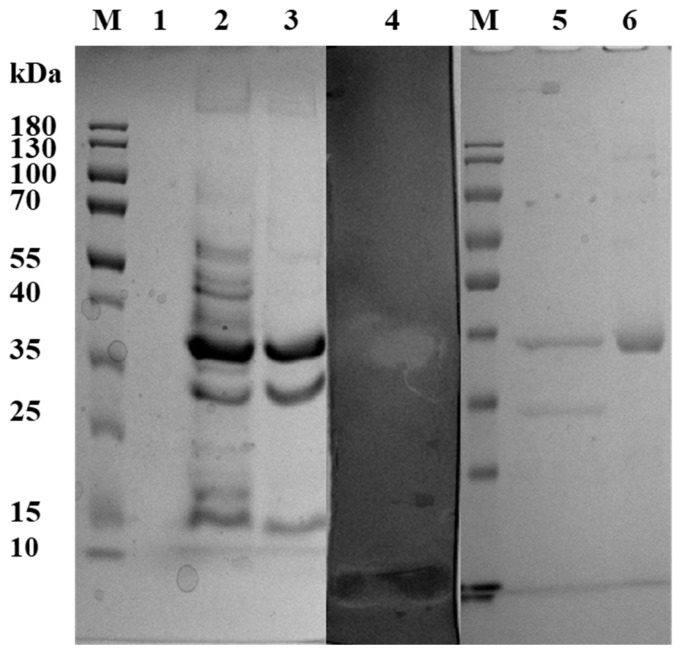
Purification and zymogram analysis of *Bp*CSN. Lane M: Molecular mass of standard proteins; Lane 1: the fermentation medium; Lane 2: crude enzyme (1.0% insoluble chitosan inducer); Lane 3: crude enzyme (2.0% insoluble chitosan inducer); Lane 4: illustrates zymogram analysis of the purified enzyme and indicates chitosanase activity; Lane 5: purified enzyme (2.0% insoluble chitosan inducer); Lane 6: purified enzyme.

**Figure 4 foods-12-03350-f004:**
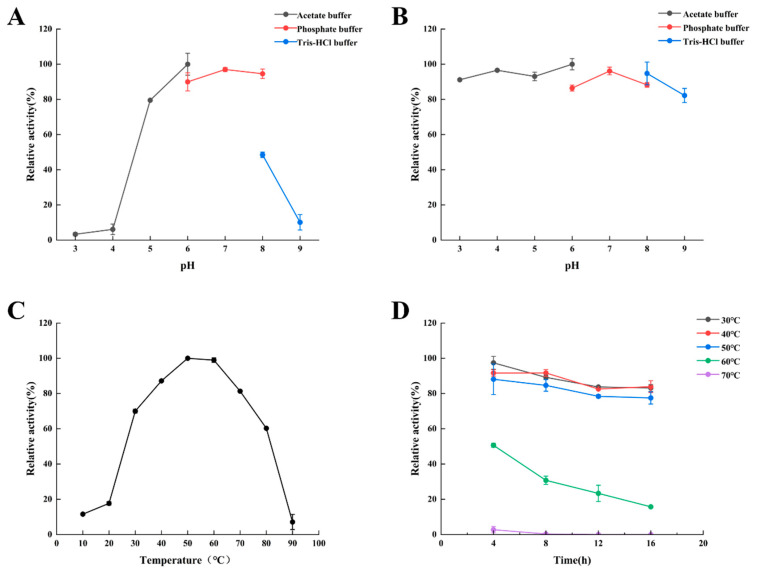
Determining the enzymatic properties of chitosanase. (**A**) The effect of pH on the activity of *Bp*CSN. (**B**) The effect of pH on the stability of *Bp*CSN. (**C**) The effect of temperature on the activity of *Bp*CSN. (**D**) The effect of temperature on the stability of *Bp*CSN.

**Figure 5 foods-12-03350-f005:**
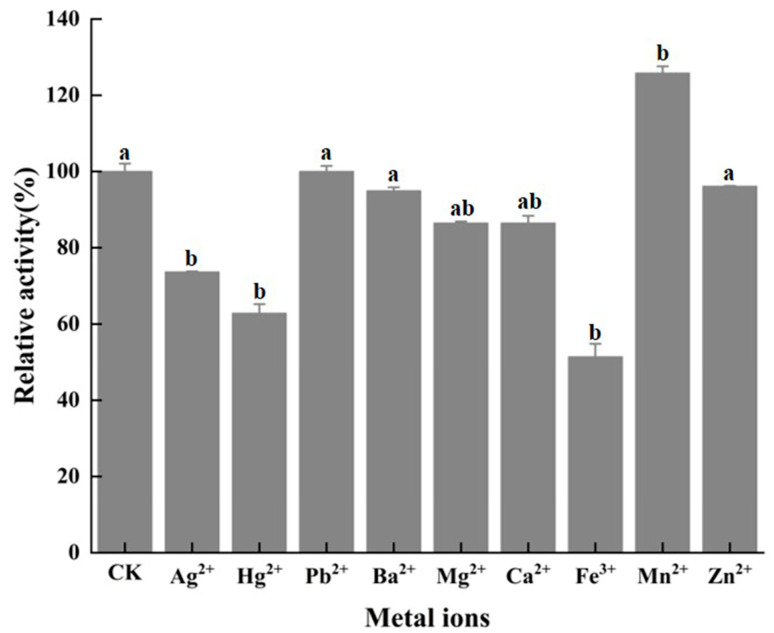
The effects of metal ions on *Bp*CSN activity. The letters “a,b” are subsets of the Duncan algorithm. The same letters indicate unsignificant differences (*p* > 0.05); different letters indicate significant differences (*p* < 0.05).

**Figure 6 foods-12-03350-f006:**
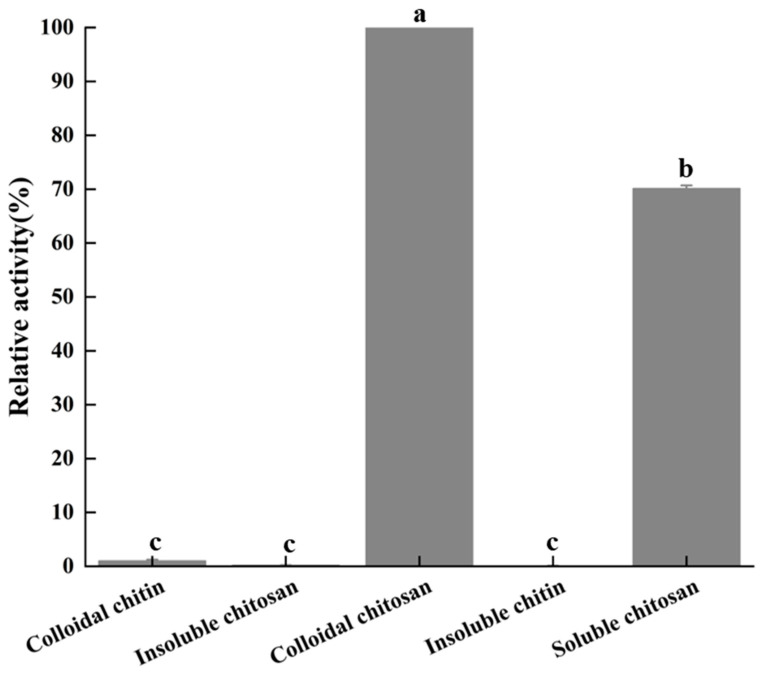
Substrate specificity of chitosanase. The relative activity was expressed as the percentage ratio of the activity of *Bp*CSN using various substrates compared to that using colloidal chitosan (DD ≥ 98.0%). The letters “a,b,c” are subsets of the Duncan algorithm. The same letters indicate unsignificant differences (*p* > 0.05); different letters indicate significant differences (*p* < 0.05).

**Figure 7 foods-12-03350-f007:**
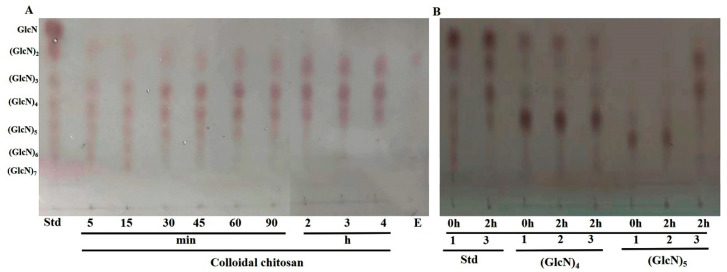
Thin-layer chromatography (TLC) analysis of hydrolytic products of chitosan catalyzed by *Bp*CSN over time. (**A**) Hydrolysis properties of chitosanase. Std: COSs mixture (1–7). E: inactivated enzymes. (**B**) TLC analysis of degradation products of colloidal chitosan. Std-1, 3: COSs Mixture after 0 h and 2 h of treatment with *Bp*CSN. (GlcN)_4_ -1: standard sample of chitotetraose. (GlcN)_4_-2: control group (chitotetraose + water). (GlcN)_4_-3: standard sample of chitotetraose after 2 h of treatment with *Bp*CSN. (GlcN)_5_-1: standard sample of chitopentaose. (GlcN)_5_-2: control group (chitopentaose + water). (GlcN)_5_-3: standard sample of chitopentaose after 2 h of treatment with *Bp*CSN.

## Data Availability

The data used to support the findings of this study can be made available by the corresponding author upon request.

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
