# Peer review of "Production, Characterization and Application of a Novel Chitosanase from Marine Bacterium Bacillus paramycoides BP-N07"

_foods, 2023, doi:10.3390/foods12183350_

Round 1

Reviewer 1 Report

Comments and Suggestions for Authors

Reviewer report on Manuscript Draft ‘Production, characterization and application of a novel chitosanase from marine bacterium Bacillus paramycoides BP-N07’

In this manuscript authors screened a chitosanase producing marine strain from sea mud samples, and the production and partial characterization of chitosanase BpCSN from a newly isolated Bacillus paramycoides BP-N07 were evaluated, which will play a positive role of chitosan industrial technology upgrade and quality improvement.

This manuscript is very interesting, from the point of view of biotechnology and biochemistry. The review is well addressed; it is in the scope of the journal. Therefore, the manuscript can be published after some minor corrections and improvements:

Characterization and application aspects of here reported chitosanase could be discussed more intensively taking into account literature on application of enzymes in various biosensorics-based applications in biosensors and other biotechnological devices (Charge transfer and biocompatibility aspects in conducting polymers based enzymatic biosensors and biofuel cells. Nanomaterials 2021, 11, 371.).

Comments on the Quality of English Language

 Minor editing of English language required.

Author Response

We are very grateful for your insightful and constructive comments on our manuscript (foods-2568297). We revised the manuscript following your advice. Modified portions are marked in red on the paper. Here below is a one-by-one response to your comments.

Reviewers' comments as follows:

Reviewer #1: Reviewer report on Manuscript Draft ‘Production, characterization and application of a novel chitosanase from marine bacterium Bacillus paramycoides BP-N07’In this manuscript authors screened a chitosanase producing marine strain from sea mud samples, and the production and partial characterization of chitosanase BpCSN from a newly isolated Bacillus paramycoides BP-N07 were evaluated, which will play a positive role of chitosan industrial technology upgrade and quality improvement.This manuscript is very interesting, from the point of view of biotechnology and biochemistry. The review is well addressed; it is in the scope of the journal. Therefore, the manuscript can be published after some minor corrections and improvements.

Thank you very much for your comments. We have carefully revised our manuscript to address the two issues. We appreciate your time and effort in helping us improve the manuscript and look forward to your final decision on publication.

(1) Characterization and application aspects of here reported chitosanase could be discussed more intensively taking into account literature on application of enzymes in various biosensorics-based applications in biosensors and other biotechnological devices (Charge transfer and biocompatibility aspects in conducting polymers based enzymatic biosensors and biofuel cells. Nanomaterials 2021, 11, 371.).

Response: Thanks very much for taking your time to review this manuscript. This is an excellent comment, we have added some sentences and relevant literatures to discuss more intensively the applications of chitosanases in biosensors and other biotechnological devices.

(2) Minor editing of English language required.

Response: Thanks for your suggestion. According to your suggestions, we have revised these grammatical, vocabulary and syntactic errors in text. Meanwhiles, we have read the text carefully and improved the English writing. For the details, see the text, please.

Reviewer 2 Report

Comments and Suggestions for Authors

This paper describes the production, characterization, and application of a novel chitosane from marine bacterium Bacullus paramycoides BP-N07. The article is quite complete, it is of interest to the scientific community, the methods and statistics used are appropriate and the results and discussion are conveniently described. The work is well discussed and is supported by the references provided by the authors. The English language is correct. The authors have a great knowledge of the subject as it is observed in the bibliography, and this work deepens even more in this field. The work is interesting and delves in the production and application of chitosane.

I consider that the article is appropriate to be published in Foods journal once the authors have made some modifications to it.

-          Title: Capitalize each word according the format of the journal.

-          Lines 36, 38, 129, 249, …..: Put a separation after and before “±”, “=”, “<”. Unify and apply to the entire document.

-          Lines 38, 134, 161, 162, 165,……: Use “mL” instead of “ml”. Unify and apply to the entire document.

-          Lines 127, 253, 270, 305,…..: Capitalize each word according the format of the journal. Unify and apply to the entire document.

-          Lines 161, 309, ……: Put a separation between the number and the units. Unify and apply to the entire document.

-          Section 2.3. Determination of Chitosane Activity: Include calibration curve, r2 and range of linearity of the calibration curve.

-          Lines 199, 204, 205, 212,….: Put a separation between a number and “ºC”. Unify and apply to the entire document.

-          Line 249: Put “n” in italics. Unify and apply to the entire document.

-          Lines 274, 309,….: Use “Figure” instead “Fig”. Unify and apply to the entire document.

-          References: Unify the format of the references: In the name of the journals, after each abbreviated word, use a dot, according the format of the journal.

-          Reference 4, 18: Information is missing.

Comments on the Quality of English Language

Correct

Author Response

We are very grateful for your insightful and constructive comments on our manuscript (foods-2568297). We revised the manuscript following your advice. Modified portions are marked in red on the paper. Here below is a one-by-one response to your comments.

Reviewer #2: This paper describes the production, characterization, and application of a novel chitosane from marine bacterium Bacullus paramycoides BP-N07. The article is quite complete, it is of interest to the scientific community, the methods and statistics used are appropriate and the results and discussion are conveniently described. The work is well discussed and is supported by the references provided by the authors. The English language is correct. The authors have a great knowledge of the subject as it is observed in the bibliography, and this work deepens even more in this field. The work is interesting and delves in the production and application of chitosane. I consider that the article is appropriate to be published in Foods journal once the authors have made some modifications to it.

We acknowledge the efforts of you and your valuable comments that improved our manuscript. We have revised the manuscript very carefully according to your comments one by one. We sincerely wish you would be satisfied with these revisions and responses and look forward to your final decision on publication.

(1) Title: Capitalize each word according the format of the journal.

Response: Sorry for my carelessness. We have capitalized the first letter of each word in the title according the format of the journal.

(2) Lines 36, 38, 129, 249, …..: Put a separation after and before “±”, “=”, “<”. Unify and apply to the entire document.

Response: Thanks for your suggestion. According to your suggestions, we have added a separation after and before “±”, “=”, “<” and applied to the entire document.

(3) Lines 38, 134, 161, 162, 165,……: Use “mL” instead of “ml”. Unify and apply to the entire document.

Response: Thanks for pointing this out. We have modified “ml” to “mL” and applied to the entire document.

(4) Lines 127, 253, 270, 305,…..: Capitalize each word according the format of the journal. Unify and apply to the entire document

Response: Thanks. According to the journal standards of Foods, we have capitalized the first letter of each word and applied to the entire document.

(5) Lines 161, 309, ……: Put a separation between the number and the units. Unify and apply to the entire document.

Response: Thanks. We have read the entire document carefully and added a separation between the number and the units.

(6) Section 2.3. Determination of Chitosane Activity: Include calibration curve, r2 and range of linearity of the calibration curve.

Response: Thanks for your suggestion. We have added calibration curve, r2 and range of linearity of the calibration curve in the Supplementary Data. The details are shown in Figure S1 below.

Figure S1. The standard curve of glucosamine (0.1-1.0 mg/mL) for the determination of chitosanase activity.

(7) Lines 199, 204, 205, 212,….: Put a separation between a number and “ºC”. Unify and apply to the entire document.

Response: Sorry for my carelessness. According to your suggestions, we have added a separation between a number and “ºC” and applied to the entire document

(8) Line 249: Put “n” in italics. Unify and apply to the entire document.

Response: Thanks. We modified “n” in italics and applied to the entire document.

(9) Lines 274, 309,….: Use “Figure” instead “Fig”. Unify and apply to the entire document.

Response: Thanks. We have modified“Fig” to “Figure” and applied to the entire document.

(10) References: Unify the format of the references: In the name of the journals, after each abbreviated word, use a dot, according the format of the journal. Reference 4, 18: Information is missing.

Response: Thanks for your suggestion. According to the format of the references of Foods, we have added a dot after each abbreviated word in the name of the journals and some incorrect references have been emended. For the details, see the text, please.